# Neurogranin Regulates Adult-Born Olfactory Granule Cell Spine Density and Odor-Reward Associative Memory in Mice

**DOI:** 10.3390/ijms22084269

**Published:** 2021-04-20

**Authors:** Simona Gribaudo, Daniele Saraulli, Giulia Nato, Sara Bonzano, Giovanna Gambarotta, Federico Luzzati, Marco Costanzi, Paolo Peretto, Serena Bovetti, Silvia De Marchis

**Affiliations:** 1Department of Life Sciences and Systems Biology (DBIOS), University of Torino, 10123 Turin, Italy; simona.gribaudo@inserm.fr (S.G.); giulia.nato@unito.it (G.N.); sara.bonzano@unito.it (S.B.); federico.luzzati@unito.it (F.L.); paolo.peretto@unito.it (P.P.); 2Institute of Cell Biology and Neurobiology (IBCN), National Research Council, 00143 Rome, Italy; d.saraulli1@lumsa.it; 3Neuroscience Institute Cavalieri Ottolenghi (NICO), Orbassano, 10043 Turin, Italy; giovanna.gambarotta@unito.it; 4Department of Clinical and Biological Sciences (DSCB), University of Torino, 10043 Turin, Italy; 5Department of Human Sciences, LUMSA University, 00193 Rome, Italy; m.costanzi@lumsa.it

**Keywords:** neurogranin, learning, calmodulin, activity-dependent plasticity, olfactory granule cells, adult neurogenesis

## Abstract

Neurogranin (Ng) is a brain-specific postsynaptic protein, whose role in modulating Ca^2+^/calmodulin signaling in glutamatergic neurons has been linked to enhancement in synaptic plasticity and cognitive functions. Accordingly, Ng knock-out (Ng-ko) mice display hippocampal-dependent learning and memory impairments associated with a deficit in long-term potentiation induction. In the adult olfactory bulb (OB), Ng is expressed by a large population of GABAergic granule cells (GCs) that are continuously generated during adult life, undergo high synaptic remodeling in response to the sensory context, and play a key role in odor processing. However, the possible implication of Ng in OB plasticity and function is yet to be investigated. Here, we show that Ng expression in the OB is associated with the mature state of adult-born GCs, where its active-phosphorylated form is concentrated at post-synaptic sites. Constitutive loss of Ng in Ng-ko mice resulted in defective spine density in adult-born GCs, while their survival remained unaltered. Moreover, Ng-ko mice show an impaired odor-reward associative memory coupled with reduced expression of the activity-dependent transcription factor Zif268 in olfactory GCs. Overall, our data support a role for Ng in the molecular mechanisms underlying GC plasticity and the formation of olfactory associative memory.

## 1. Introduction

The olfactory bulb (OB) is the first relay station of sensory stimuli coming from the olfactory epithelium in the nose. OB mitral/tufted cells (M/TC) receive inputs from the olfactory axons in the glomerular layer and project to the olfactory cortex. The activity of M/TC is influenced by a variety of local OB interneurons which modulate sensory processing through synaptic inhibition [1]. Among inhibitory interneurons, granule cells (GCs), forming the GC layer (GCL) in the OB, are the most abundant. GCs form reciprocal dendrodendritic synapses with M/TC via large spines on their distal dendrites in the external plexiform layer (EPL), and receive synaptic top-down inputs from the axons of centrifugal fibers originating in olfactory cortical areas and targeting the GCL [2,3]. Functionally, GCs regulate firing synchronization of M/TC allowing a fine spatio-temporal tuning of their responses to odorants [1,4,5,6]. Importantly, olfactory processing in the OB exhibits significant plasticity as a result of previous odor experience and the persistent addition of newly generated neurons through adult neurogenesis [7,8]. In this context, adult-born GCs play an active role in experience-dependent plasticity, thanks to their high structural plasticity and synaptic remodeling in response to the sensory context [9,10,11,12,13]. 

We have previously shown that a large subset of GABAergic GCs, mainly located in the deep GCL (dGCL), express the postsynaptic protein Neurogranin (Ng) [14]. Ng is a calmodulin (CaM)-binding protein that is phosphorylated by protein kinase C (PKC) and is highly enriched in the dendrites and spines of telencephalic glutammatergic neurons, where it controls CaM availability and distribution at post-synaptic sites [15,16,17,18,19,20]. In the adult hippocampus, where Ng has been deeply investigated, its function in modulating Ca^2+^/CaM signaling causes an enhancement in synaptic plasticity [19,21,22,23,24,25,26,27]. Accordingly, Ng knock-out (Ng-ko) mice show severe deficits in hippocampal-dependent spatial learning associated with impairment in long-term potentiation (LTP) induction and decreased activation of Ca^2+^/CaM-dependent signaling [22,24,28]. Moreover, a recent study demonstrated that Ng expression is upregulated in the hippocampus of mice following a contextual memory test and its activity-dependent increase is required for contextual memory formation [29]. Importantly, Ng dysregulation is associated with several diseases including schizophrenia [30,31], Jacobsen syndrome—a rare genetic disorder with symptoms of intellectual disability [32]—and Alzheimer’s disease [33,34,35,36,37]. In animal models (i.e., 5XFAD mice) and patients with Alzheimer’s disease, Ng is downregulated in frontal cortex and hippocampus, and viral-mediated restoration of its levels in the hippocampus of 5XFAD mice contributes to the enhancement of cognitive functions [38], further demonstrating a strong correlation between brain Ng expression and cognitive performances.

The importance of Ng in synaptic plasticity and its expression in olfactory GABAergic GCs suggest Ng may be involved in regulating the high structural and functional plasticity of OB circuits. In this study, we show that Ng expression in adult-born GCs correlates with their maturation and functional integration within OB circuits. In Ng-ko mice, the lack of Ng does not affect the survival of adult-born GCs, but alters their dendritic spine density, mainly within the GCL. Moreover, analysis of the immediate early gene Zif268 [39], whose expression in the GCL is modulated by experience [40], shows reduced activation in olfactory GCs coupled to impaired odor-reward associative memory in Ng-ko mice, providing evidence for Ng function in OB experience-dependent plasticity. 

## 2. Results

### 2.1. Subcellular Localization of Ng and Its Phosphorylated form in Olfactory GC Interneurons

At a cellular level, Ng-immunostaining in the OB was observed in the somato-dendritic compartment of GCs (Appendix A), which co-express the Ca^2+^/CaM-dependent CamKIIα (Figure 1b–d) and/or CamKIV [14]. In the external plexiform layer (EPL), where the distal dendrites of GCs extensively ramify to form reciprocal dendro-dendritic synapses with mitral/tufted cells (M/TCs) [41], Ng-labelling showed a pattern complementary to that of the presynaptic marker synaptophysin (Syp; Figure 1e,f), consistent with Ng localization at postsynaptic sites [42,43]. Interestingly, western blot analysis on cellular fractions obtained from OB samples revealed an Ng protein signal in the cytoplasmic (Cyt) and synaptosomal (Syn) enriched fractions, while the Ng phosphorylated form (pNg) was mostly concentrated in the latter, which also selectively expressed the post-synaptic density protein 95 (PSD-95; Figure 1g). Ng phosphorylation by protein kinase C (PKC) at a serine (Ser36) within the CaM-binding motif prevents CaM binding, facilitating Ca^2+^/CaM signaling and synaptic plasticity [19,20,27]. Thus, our findings suggest Ng could be involved in olfactory GC synaptic plasticity.

### 2.2. Ng Expression in Adult Born GCs Is Associated with Their Mature State and Is Dispensable for Their Survival

Adult born olfactory GCs play a pivotal role in synaptic remodeling and odor information processing [9,10,11,12,13]. We therefore focused on the expression and role of Ng in adult-generated GCs. Newborn cells were labelled with BrdU and Ng immunoreactivity was analyzed at different survival times post-BrdU injection in adult mice (Figure 2a). Two weeks post-injection, when newborn GCs are still immature [44], only about 8% of BrdU-positive GCs co-expressed Ng. The percentage of Ng-positive cells drastically increased to about 40% by 21 days, when most newborn GCs have developed spines on apical processes [44]. The percentage of BrdU-positive cells co-expressing Ng remained at similar levels in fully mature GCs at 42- and 63-days survival (Figure 2a).

We next addressed the effect of Ng loss or reduction on the survival of newborn GCs by taking advantage of Ng-ko and Ng-heterozygous (Ng-het) mouse models [22]. In Ng-ko mice, we did not detect Ng immunostaining, and the lack or reduction of Ng protein signal was confirmed by western blot in the OB of Ng-ko or Ng-het mice, respectively (Appendix A). The density of BrdU-positive GCs in the OB of Ng-ko mice was quantified and compared to that of Ng-het and wt mice at 42 days survival (Figure 2b,c). Since the density of Ng-positive GCs is higher in the deeper compared to the superficial GCL [14], we analyzed the two subregions of the GCL independently (Figure 2b; https://github.com/olivierfriard/NCSD, accessed 25 October 2019) [45]. The density of BrdU-positive GCs was similar in all genotypes for both subregions (Figure 2c). Moreover, no differences were observed in the volume of the different OB layers in Ng-ko compared to wt and Ng-het animals (Appendix A). Overall, these data indicate that a constitutive lack or reduction of Ng expression has no impact on the gross morphology of the OB and does not alter the survival of adult-generated GCs.

### 2.3. Defective Spine Density in Olfactory Adult-Born GCs of Ng-ko Mice

To further address whether the lack of Ng influences adult generated GCs, we labelled SVZ-progenitors in wt and Ng-ko mice through intraventricular injection of a retrovirus encoding the green fluorescent protein (GFP) [46,47] and performed morphometric analyses on the dendrites of GFP-positive GCs at 6 weeks post-injection (Figure 3 and Figure 4). We restricted the analyses to the deep GCL, which in wt mice is highly enriched by Ng-positive cells [14] (Figure 1a,b). Only cells showing a high level of GFP expression with basal and/or apical processes entirely included in the analyzed volume (120 µm) were processed for analysis (Figure 3a). The average length of the apical dendritic tree did not differ in Ng-ko versus wt mice (Figure 3b). However, Sholl analysis revealed a decreased probability of ramification between 150 and 200 μm in cells from Ng-ko mice, whereas the number of intersections was increased at longer distances from the cell soma (between 350 and 380 μm; Figure 3c). The total length of basal dendrites was slightly higher in Ng-ko compared to wt mice (Figure 3d). However, no difference was found in the number of branches and bifurcations between the two groups in the basal compartment (Figure 3e,f), as further confirmed by Sholl analysis (Figure 3g).

We next evaluated the density and length of the dendritic spines in both the distal part of the apical dendrites and in the basal dendrites of wt and Ng-ko mice (Figure 4). Interestingly, a net reduction in spine density was found in the basal GCs dendritic compartment of Ng-ko mice (Figure 4f), while no statistically significant difference was observed in the apical compartment (Figure 4e). In addition, no difference was observed in spine length between genotypes in both the apical and basal dendritic compartments (Figure 4g–j).

Overall, these data indicate alterations in the morphology of adult-generated GCs in Ng-ko mice, including moderate changes in the apical dendrite ramification pattern, and a slight increase in the total basal dendritic length associated with a robust decrease in spine density in the basal compartment.

### 2.4. Ng-ko Mice Show Impaired Odor-Reward Associative Memory Associated to Defective Zif268 Expression in Gcs

To investigate possible alterations in Ng-ko mice olfactory function, we initially tested the ability of Ng-ko mice to detect and discriminate octanal and acetophenone, which are two structurally and perceptually different odorants. To this aim, we exposed wt and Ng-ko mice to a series of progressively increasing concentrations of the two odorants (in separate experimental sessions), and measured the time they spent investigating such odors versus odorless mineral oil (Figure 5a). A significant increase in the time spent investigating the odor compared to mineral oil was considered as an indication of the animal’s odor detection capacity [9]. For both octanal and acetophenone, wt and Ng-ko mice were able to detect the odor at a concentration equal to or greater than 10^−4^ % with no statistically significant differences between groups in the odor preference index (Figure 5a). Next, their ability to discriminate between octanal and acetophenone was assessed by a habituation-dishabituation test, consisting in four habituation (Hab) trials wherein animals were repeatedly exposed to octanal, followed by one dishabituation (Dishab) trial, in which they were exposed to the novel odor acetophenone (Figure 5b) [48]. In both wt and Ng-ko mice, the time spent investigating octanal during the habituation phase declined gradually. Conversely, it significantly increased during the dishabituation phase, with no statistically significant differences between the two genotypes, indicating a comparable ability of wt and Ng-ko mice to discriminate between such odors (Figure 5b). Altogether, these data indicate that Ng-ko mice have a normal ability to detect these two odorants, and to discriminate efficiently between them. We then assessed the performance of Ng-ko mice in an odor-reward associative memory test [49]. To this end, we trained mice for four consecutive days to associate octanal with a sugar reward hidden beneath the surface of the wood chip bedding (Figure 6a, top). On day five, the memory for the odor-reward association was evaluated by measuring the time the animals spent digging at octanal, compared to the time they spent digging at acetophenone, which was never associated to reward during the training (Figure 6a, bottom). In this test, wt mice demonstrated a strong memory for the reinforced odor (O+), by digging at octanal, nearly four times longer than they did for acetophenone (A-), while Ng-ko mice did not show any significant preference for the reinforced odor (O+), compared to the unreinforced (A-) (Figure 6b). Moreover, the latency to dig at octanal (i.e., the time elapsed from the beginning of the test to the first episode of digging) was significantly higher for Ng-ko mice compared to wt (Figure 6c). These data indicate a substantial impairment of olfactory-mediated associative memory in Ng-ko mice, possibly linked to aberrant GC activation [50,51]. To verify this possibility, at the end of the behavioral tests, mice were euthanized and processed for immunostaining for the early immediate gene product Zif268, used as a proxy for neuronal activity (Figure 6d,e) [39,40]. Quantification of Zif268 immuno-positive nuclei in the GCL revealed a statistically significant reduction in Ng-ko compared to wt mice, in both the superficial and deep GCL (Figure 6e). Interestingly, the strongest decrease occurred in the deep GCL (50% reduction in dGCL versus 25% reduction in superficial GCL), where higher density of Ng expressing GCs are present in wt mice [14]. These findings support a role for Ng in GC interneuron recruitment in the context of olfactory-mediated associative memory.

## 3. Discussion

Ng is a neuronal protein whose role in synaptic plasticity, learning, and memory functions has been thoroughly investigated in forebrain glutamatergic neurons [20,22,27,29,52]. We have previously demonstrated that in the adult mouse OB, Ng is selectively expressed in a large subpopulation of GABAergic GC interneurons [14]. Our current data extend these observations by showing that the Ng phosphorylated form, which is essential for the activation of the Ca^2+^/CaM-dependent pathway and refinement of synaptic plasticity [53], was concentrated in the synaptosome-enriched fraction obtained from OB tissue extracts, suggesting a role for Ng in GC synaptic plasticity and its possible implication in olfactory memory. Accordingly, our results demonstrate that loss of Ng function in Ng-ko mice induces an impairment in odor-reward associative memory, without affecting odor detection and discrimination. Using the same mouse model, previous studies have shown that Ng loss-of-function causes impaired spatial learning and changes in short- and long-term plasticity in hippocampal neurons [22,23] and highlighted a significant correlation between learning performance and Ng levels in Ng-heterozygous mice [24]. This positive correlation is further supported by the demonstration that increasing Ng expression at dendritic spines facilitates LTP and enhances plasticity and extinction learning by targeting CaM localization within dendritic spines, promoting activation of Ca^2+^/CaM downstream signaling [19,20,27,52,54,55]. Thus, the lack of Ng in olfactory GCs might cause impaired targeting of CaM to dendritic spines and have an impact on downstream signaling events (e.g., CamKII and CamKIV activation) which are essential for memory encoding [56]. Accordingly, it has been shown that OB infusion of the kinase inhibitor K252a, which blocks, among others, the PKC and both CamKII and CamKIV, inhibits the consolidation of incremental, appetitive odor memory [57].

Olfactory GC circuitry is considered dynamic and adaptive and can be modified by various factors including sensory experience, learning, and memory [9,10,11,12,13]. The continuous production of GC interneurons contributes to olfactory plasticity and previous studies have shown that adult-born GCs display structural and functional plasticity of their inputs as well as of their output synapses in response to different behavioral paradigms, including olfactory learning and memory [58,59]. In addition, exposure to reward-associated odors specifically increases the activity of adult-born neurons [60]. Interestingly, we found that defective odor memory in Ng-ko mice is associated with a reduction in the expression of the immediate early gene Zif268 in the GCs, particularly in the deep GCL, where most of Ng-positive cells are located in wt mice [14], indicating an impairment in GC interneuron activation.

Ng expression in adult-born GCs correlates with GC synaptic maturation and their integration within OB circuits and peaks 3 weeks after birth, when newborn GCs reach the highest degree of synaptic plasticity [44,61,62,63]. Our findings show that the survival of adult-born GCs in Ng-ko mice is unaltered and morphological analysis of these cells indicates no major defects in the apical dendritic compartment, but a net decrease in spine density in the basal dendritic domain. In line with a role for Ng in sensory-dependent synaptic plasticity, a study on the mouse primary visual cortex during the critical period demonstrated that Ng loss decreases AMPAR-positive synapse number, prevents AMPAR-silent synapse maturation, and increases spine elimination [64]. Intriguingly, a similar mechanism could be involved in olfactory-experience-dependent plasticity, since AMPARs have been shown to promote spine maintenance in adult-born GCs and to reinforce synaptically active spines [65]. Moreover, the basal dendritic region of adult-born GCs is highly responsive to olfactory learning and is directly targeted by top-down excitatory drive from the olfactory cortices [58]. Thus, the observed decrease in spine density selectively in this region suggests reduced excitatory inputs from the olfactory cortex onto newborn GCs, in line with the observed impairment in olfactory learning. Furthermore, since Ng expression in the brain includes regions involved in olfactory memory circuits [58,66], its loss in Ng-ko mice could impair their function and contribute to the observed phenotype. Further experiments are needed to selectively delete Ng in olfactory GCs in order to isolate its direct involvement in GCs to mediate odor-reward associative memory function.

In conclusion, our data show that Ng-ko mice have reduced activation of olfactory GCs during an odor-reward associative task, impaired olfactory associative learning, and defective spine density in the GC basal domain. These data, together with the preferential expression of Ng in a large subset of olfactory GCs in the deep GCL, points to a direct involvement of Ng function in OB experience-dependent plasticity and strengthen the hypothesis that specific GC subtypes play distinct roles in olfactory processes [67], suggesting that the Ng-expressing GC subpopulation may play a unique role in the formation of olfactory associative memory.

## 4. Materials and Methods

### 4.1. Animals and Housing Conditions

All experimental procedures were conducted in accordance with the Guide for the Care and Use of Laboratory Animals of the European Community Council Directives (2010/63/EU and 86/609/EEC) and approved by the local Bioethics Committees and the Italian Ministry of Health. All animals were housed under a 12-h light:dark cycle in an environmentally controlled room with a maximum of 5 animals per cage. Experiments were performed on adult mice (8 weeks old, either sex). Both C57BL/6 J mice (Charles River, Calco, Italy) and Neurogranin knock-out (Ng-ko), heterozygous (Ng-het) and wild type (wt) littermates as control (B6.129-Ngrtm1Kph/J, JAX stock #008233, Jackson Laboratory; Bar Harbor, ME, USA) were used. For a detailed description of the generation and characterization of Ng-ko mice, see [22]. Genotyping was performed following the instructions specified by the Jacksons laboratory and available online. All behavioral experiments were performed during the second half of the lighting period (between 2:00 and 5:00 p.m.) in acoustically insulated rooms.

### 4.2. BrdU Injection

Eight-week-old mice received four intraperitoneal injections (4 h apart, on the same day) of 5-bromo-2-deoxyuridine (BrdU; 50 mg/kg in 0.1M Tris pH 7.4; Sigma Aldrich, St. Louis, MO, USA). Mice were left to survive for 10, 15, 21, 42, and 63 days after BrdU administration.

### 4.3. Surgical Procedure and Retroviral Injection

Eight-week-old mice were anesthetized by an intraperitoneal injection of a mixture of ketamine (100 mg/kg body weight, Ketavet; Gellini, Aprilia LT, Italy) and xylazine (10 mg/kg body weight, Rompun; Bayer, Wuppertal, Germany), positioned in a stereotaxic apparatus (Stoelting, Wood Dale, IL, USA), and maintained on a warm platform at 35 °C to keep body temperature constant during anesthesia. The skull was exposed by an incision in the scalp and a small hole (about 1 mm) drilled through the skull. Injection of 1 µL of the retroviral vector (CAG-GFP; titer 0.5 × 10^7^ cfu mL^−1^ [46,47]; generous gift from Dr. Chichung Lie; Friedrich-Alexander-Universität Erlangen-Nürnberg; Erlangen, Germany) was made at stereotaxic coordinates of 1 mm anterior to bregma, 0.8 mm lateral to the sagittal sinus, and a depth of 3 mm, using a glass micropipette and a pneumatic pressure injection apparatus (Picospritzer II, General Valve Corp., Fairfield, IL, USA). Animals were then placed into breeding standard cages and monitored until they resumed feeding and grooming activity and were euthanized 6 weeks after viral injection. Brain tissue was processed and analyzed by immunofluorescence as described below.

### 4.4. Tissue Preparation and Sectioning

Animals were deeply anesthetized by an intraperitoneal injection of a mixture of ketamine (150 mg/kg body weight, Ketavet; Gellini, Aprilia LT, Italy) and xylazine (20 mg/kg body weight, Rompun; Bayer, Wuppertal, Germany) and perfused transcardially with 0.9% saline, followed by 4% paraformaldehyde in 0.1 M phosphate buffer, pH 7.4. Brains were removed from the skull, postfixed for 4–6 h in the same solution, cryoprotected in a 30% sucrose solution in 0.1 M phosphate buffer, pH 7.4, frozen, and cryostat sectioned (Leica Microsystems, Milan, Italy). Free floating coronal serial sections (25 or 40 μm in thickness) were collected in multiwell dishes. Sections were stored at −20 °C in antifreeze solution until use.

### 4.5. Immunohistochemistry

Sections were incubated overnight at 4 °C in primary antibodies diluted in 0.01 M PBS, pH 7.4, 0.5% Triton X-100, and 1% normal serum of the same species of secondary antiserum. The primary antibodies used were anti-BrdU (1:3000; rat monoclonal, AbD Serotec, Bio-Rad Laboratories, Hercules, CA, USA; cat. #OBT0030CX), anti-Neurogranin (1:1000; rabbit; Chemicon; Temecula, CA, USA; cat. #AB5620), anti-GFP (1:1000; Chicken, polyclonal antibody, Aves Lab; Tigard, OR, USA; cat. #GFP-1020), anti-calretinin (1:2000; mouse, Swant; Marly, Switzerland; cat. #6B3), anti-synaptophysin (1:500; mouse, Sigma Aldrich; St. Louis, MO, USA; cat. #S5768), anti-Zif268 (1:1000; rabbit, Santa Cruz Biotechnology; Dallas, TX, USA; cat. #SC189), anti-CaMKIIα (1:500, mouse monoclonal, Chemicon; Temecula, CA, USA; cat. #MAB3119). For double labelling with BrdU, sections were first incubated overnight at 4 °C in anti-Ng primary antibody and appropriate serum, followed by 1 h incubation at room temperature in the proper secondary antibody. Sections were then processed for BrdU immunostaining following pre-treatment with 2 N HCl for 35 min at 37 °C neutralized with 0.1 M borate buffer at pH 8.5. Secondary antibodies used were as follows: anti-mouse, anti-rat, anti-rabbit Cy3 conjugated (1:800; Jackson ImmunoResearch, West Grove, PA, USA; cat. #715-165-151, #712-165-153 and #711-165-152); anti-rabbit, anti-mouse (1:250; Vector; Burlingame, CA, USA; cat. #BA-1000 and #BA-2000) followed by avidin FITC incubation (1:400; Vector; Burlingame, California, USA; cat. #A-2001); anti-chicken AlexaFluor488 conjugated (1:400; Jackson ImmunoResearch, West Grove, PA, USA; #703-545-155). Sections were counterstained with DAPI, mounted, air dried, and cover slipped in polyvinyl alcohol with diazabicyclo-octane (DABCO) as an antifading agent.

### 4.6. Protein Fractionation and Western Blot

Mechanical fractionation of two-to-three olfactory bulbs was performed in a solution containing 0.32 M sucrose, 1 mM NaHCO_3_, 1 mM EGTA, 1 mM MgCl_2_, 0.5 mM CaCl_2_, 1 mM DTT (buffer A) and a small amount of protein extract was kept as total homogenate (H). Samples were then centrifuged 10 min at 1400× *g*, 4 °C. The supernatant (S1 fraction) was removed and the pellet resuspended in buffer A and further centrifuged 10 min at 710× *g*, 4 °C, to obtain the S1′ supernatant and the nuclear fraction enriched pellet (NF). S1 and S1′ supernatants were combined and centrifuged 10 min at 13,800× *g* to obtain the S2 supernatant (cytoplasmic enriched fraction). The pellet obtained from the last centrifugation was resuspended in buffer A and stored as the crude synaptosomal enriched fraction.

Protein quantification was performed by a bicinchoninic acid kit (Sigma Aldrich; St. Louis, MO, USA). Briefly, 20 µg of total proteins were denatured by boiling in Laemmli buffer (2% SDS, 50 mM Tris-HCl, pH 7.4, 20% β-mercaptoethanol, and 20% glycerol) and subjected to 15% SDS-PAGE. Proteins were blotted onto Hybond-C Extra membrane (GE Healthcare Bio-Sciences; Uppsala, Sweden). After blocking with 5% powder milk in TBST buffer (20 mM Tris, 150 mM NaCl, and 0.1% Tween 20, Ph 7.4), filters were probed either with anti-phospho-Ng (pNg, 1:3000, rabbit, Chemicon; Temecula, CA, USA; cat. #07-430), anti-Ng (1:6000; rabbit; Chemicon; Temecula, CA, USA; cat. #AB5620), anti-PSD-95 (1:1000; UC Davis/NIH NeuroMab Facility; Dallas, CA, USA; cat. #75-028), and sequentially, with anti-β-Actin (1:10,000; mouse, Sigma Aldrich, St. Louis, MO, USA; cat. #A5316) in 1% Bovine Serum Albumine (BSA, Sigma Aldrich; St. Louis, MO, USA) in TBST buffer over/night at 4 °C. After rinsing in TBST buffer, the membranes were probed with the appropriate peroxidase-coupled anti-rabbit or anti-mouse (Abcam, Cambridge, MA, USA) secondary antibody and revealed with West-Pico enhanced chemiluminescence (ECL) detection system (Pierce Biotechnology, Rockford, IL, USA). Appropriate films (Kodak; Rochester, New York, NY, USA) were impressed by chemiluminescent bands and subsequently kept in developing and fixing solutions (Sigma Aldrich; St. Louis, MO, USA) until band detection.

### 4.7. Image Analysis and Quantification

Cell counts and image analysis were performed on a Nikon microscope coupled with a computer-assisted image analysis system (Neurolucida software, MicroBrightField, Colchester, VT, USA) or on a Leica SP5 confocal microscope (Leica Microsystems, Milan, Italy) using the 20× (NA = 0.75), the 40× (NA = 1.25) and the 63× (NA = 1.4) oil-immersion objectives. To establish the percentage of new-generated GCs double-labelled for Ng, serial sections were examined systematically through the olfactory bulbs (3 sections/animal positioned, respectively, at the anterior, medial, and posterior level). Confocal image z-stacks were captured through the thickness of the slice (1 µm optical step) and the percentage of Ng-positive newborn cells in the GCL was calculated using a random sampling method over the total number of BrdU-positive cells.

To assess the density of new-generated GCs at different survival ages in wt, het, and Ng-ko mice, all BrdU-positive cells in the GCL (3 sections/animal positioned respectively at the anterior, medial and posterior level) were counted using Neurolucida. On each slice, we traced the contour of the GCL and of the sub-ventricular zone (SVZ) of the olfactory bulb. The files were then imported in https://github.com/olivierfriard/NCSD, accessed 25 October 2019, to calculate the density of BrdU positive cells in different sub-portions of the GCL as previously described [45]. To estimate the volume of each OB layer in the different genotypes, the boundaries of each layer were manually traced based on DAPI staining and the areas of each section and layer were automatically calculated by Neurolucida software. In particular, tracing was performed on every section (i.e., eight to nine 25 µm-thick consecutive sections, with 300 µm intersection intervals) of one series per animal, and the total volume of the OB (from the anterior MOB to the anterior AOB) and layers were estimated by applying the Cavalieri method [68].

For quantification of Zif268-positive cell density, the entire GCL was acquired with a Leica SP5 confocal microscope (Leica Microsystems, Milan, Italy) using the tile-scan module and a 40× (NA = 1.25) oil-immersion objective maintaining the same acquisition parameters for wt and Ng-ko animals. A single confocal plane was acquired from 3 sections/animal positioned at the anterior, medial, and posterior anatomical levels. GCL mosaics were imported into Fiji software [69] and the boundaries of the deep and superficial GCL were drawn using DAPI and calretinin immunostaining. The number of total, deep and superficial Zif268 positive cells were automatically counted after imposing a fixed threshold and applying the watershed filter to each image. Settings were maintained the same for each image. The area of the deep, superficial, and total GCL was calculated by the software and the density expressed as cells/mm^2^.

For cell reconstruction, 1024 × 1024 images (40 µm section thickness) were taken with a Leica SP5 confocal microscope (Leica) using a 20x objective (NA = 0.75), a pinhole setting of one Airy unit and optimal settings for gain and offset. The detailed procedure has been previously described [70]. Briefly, images were acquired to cover the extension of the entire tissue slice. Images from three consecutive slices for each animal were obtained. First, the manual montage (or stitching) of multiple fields in a single mosaic image was manually performed in VIAS 2.4 [71]. Once this procedure was performed on the three slices, each slice was imported to Reconstruct [72,73] and section alignment was performed using rigid-body transformation to avoid deformation. The result is a single file containing information from 3 serial sections, accounting for a total thickness of 120 μm. Lateral and z-axis resolutions were 505 nm and 2.5 μm. The actual value of z in the voxel size was calculated a posteriori by dividing the number of optical planes by the total thickness (120 μm). Cell reconstructions of the basal and apical compartments were performed using Neuronstudio 0.9.92 [71], a freeware software specifically designed to automatically trace neuronal elements. Dendritic length, number of branches, and branch points were obtained by the software. Sholl analysis was performed by analyzing swc files obtained by Neuronstudio with the Simple Neurite Tracer plugin in ImageJ. Concentric circles spaced 5 µm for basal dendrite and 10 µm for apical dendrites were used to run the Sholl analysis. Quantifications were performed on at least 6 cells for each animal for basal dendrites and on at least 3 cells for each animal for apical dendrites.

For analysis of spine density, 1024 × 1024 images were taken with the 63× (NA = 1.4) oil-immersion objective and 3x digital zoom. Lateral and z-axis resolutions were 80 and 295 nm, respectively. Dendritic spine length was analyzed using the NeuronStudio software [74]. The number of dendritic spines was manually counted and divided by the dendrite length to get the dendritic spine linear density.

### 4.8. Odor Detection Threshold Test

The test was performed as previously described [9], with minor modifications. In a cage identical to those used to house the animals, with the floor covered by a thin layer of wood chip bedding, two tissue culture dishes (35 × 10 mm) were placed in close proximity of two opposite sides of the cage, whose covers had been previously drilled, so that they presented eight small holes each, allowing the odors to spread. Each culture dish, properly sealed with a narrow strip of Parafilm M, contained a piece of filter paper (1.5 × 1.5 cm) soaked, in one case, with odorless mineral oil (25 µL), as a control, and, in the other, with octanal, or acetophenone (25 µL), diluted in mineral oil at different concentrations (all three from Sigma-Aldrich). During four successive sessions, lasting 3 min each and separated by 15 min intervals, each mouse was exposed to culture dishes, one of which contained progressively increasing concentrations of the odorant (10^−7^, 10^−5^, 10^−4^, and 10^−3^ %). Each session was video-recorded, and the time the animals spent sniffing at the culture dishes, defined as a nasal contact with the dish within a 0.5 cm distance, was subsequently measured by using the Noldus EthoVision software (Wageningen, The Netherlands) by an experimenter blind to their genotype. For the statistical analysis, an ‘odor preference’ index was calculated as the ratio between the time spent investigating the odor and the total sniffing time (odor plus mineral oil), so that index values between 0.50 and 1.00 were indicative of a preference for the odor, compared to mineral oil.

### 4.9. Odor Discrimination

The test was performed as previously described [50], with minor modifications. In a cage identical to those used to house the animals, each mouse was familiarized with octanal (10^−3^ % in mineral oil) as the odor of habituation, during four successive trials, and then exposed to acetophenone (10^−3^ % in mineral oil) as the odor of dishabituation, in the next, final trial. Each trial lasted 3 min, and the trials were separated by 15 min intervals. Tissue culture dishes, containing filter paper soaked with the odorants, were prepared in the same manner as described for the odor detection threshold test. Each trial was video-recorded, and the time the animals spent sniffing at the culture dish, defined as a nasal contact with the dish within a 0.5 cm distance, was subsequently measured by using the Noldus EthoVision software by an experimenter blind to their genotype.

### 4.10. Odor-Reward Associative Memory

The test was performed as previously described [49] with minor modifications. Four days prior to training, animals were placed on a food restriction schedule and fed sufficiently to maintain 80–85% of their free feeding weight. During the training, animals received four 10 min trials per day, in cages identical to their home cages. For two of the trials, a tissue culture dish diffusing octanal (50 µL, 10^−3^ % in mineral oil), was hidden beneath the surface of the wood chip bedding and paired with sugar reinforcement buried into the bedding. For the remaining two trails, an identical dish, diffusing acetophenone (50 µL, 10^−3^ % in mineral oil), was hidden without sugar. The culture dishes were prepared in the same manner as described for the odor detection threshold test. The order of the trials was counterbalanced across the training. The test was performed on day five, in a larger cage (20 × 70 × 20 cm) made of transparent plexiglass and divided into three equal compartments, with openings (6 × 6 cm) leading to both end compartments. Two culture dishes, diffusing octanal (50 µL, 10^−3^ % in mineral oil) and acetophenone (50 µL, 10^−3^ % in mineral oil), were placed in the end compartments of the cage, abundantly covered by the chip bedding, with no reinforcement. The test was video-recorded, and the time the animals spent digging the bedding in close proximity of each dish was measured using the Noldus EthoVision software, during a 4 min period by an experimenter blind to their genotype. A brief habituation session, lasting 2 min, preceded the test, during which the animals were given the opportunity to explore the apparatus, with no odorants inside. After habituation, the animals were returned to their home cages for approximately 10 min prior to the start of the test.

### 4.11. Statistics

Values are expressed as mean ± SEM unless otherwise stated. A Kolmogorov-Smirnov normality test was run on each experimental sample. When comparing two populations of data, Student’s t-test was used to calculate statistical significance in case of Gaussian distribution; otherwise, the nonparametric Mann-Whitney test was used. All tests were two-tailed. When more than two populations of data were compared and in case of Gaussian distribution, either one-way or two-way repeated measures ANOVA with Fisher’s PLSD or Sidak’s post hoc test respectively was used and significance was established as follows: * *p* < 0.05; ** *p* < 0.01; *** *p* < 0.001; NS, not significant.

## Figures and Tables

**Figure 1 ijms-22-04269-f001:**
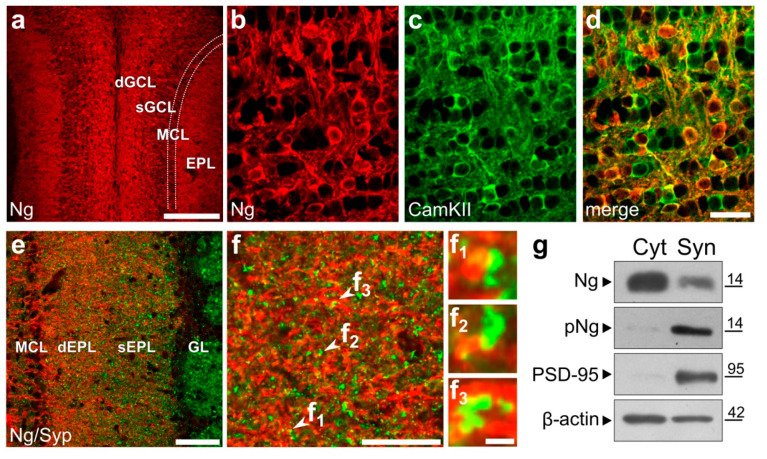
Subcellular localization of Ng and its phosphorylated form in olfactory GC interneurons. (**a**) Neurogranin immunostaining in a coronal section of the adult mouse OB labels a large population of GABAergic interneurons in the GCL, and their apical processes in the EPL. Scale bar = 200 μm. (**b**–**d**) Double labelling for Ng (**b**, red) and CamKII (**c**, green) shows high colocalization between the two markers in the deep GCL (**d**, merge). Scale bar in d = 20 μm and applies to (**b**–**c**). (**e**) Double labelling for Ng (red) and the presynaptic protein synaptophysin (Syp; green). Scale bar = 50 μm. (**f**) High magnification of the EPL stained for Ng (red) and for Syp (green). Insets (f1–f3) show the juxtaposition of Ng and synaptophysin labelling according to the post- and pre-synaptic expression of the two proteins. Scale bar in f = 20 μm; scale bar in f3 = 5 μm and applies to f1–f2. (**g**) Western blot analysis of Ng, phosho-Ng (pNg), PSD-95 and β-Actin on sub-cellular protein fractions (20 μg) obtained from whole OB samples. The band corresponding to PSD-95 is almost exclusively present in the synaptosomal fraction, indicating a reliable enrichment of the specific synaptic protein in the proper sub-cellular domain. Ng antibody detects a band in both cytoplasmic and synaptosomal enriched fractions, whereas its phosphorylated form is mainly detectable in the synaptosomal fraction. dGCL: deep granule cell layer; sGCL: superficial granule cell layer; MCL: mitral cell layer; EPL: external plexiform layer; GL: glomerular layer; Syp: synaptophysin; Cyt: cytoplasmic fraction; Syn: synaptosomal fraction.

**Figure 2 ijms-22-04269-f002:**
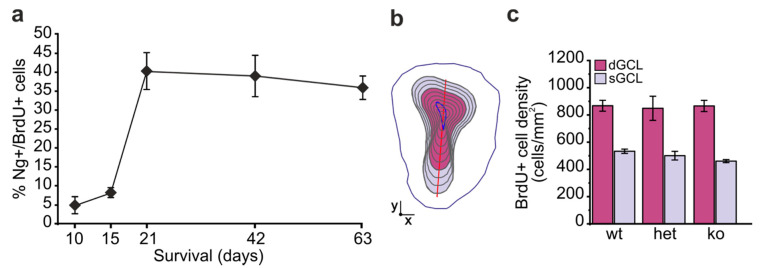
Ng expression in adult-born GCs is associated with their mature state and is dispensable for their survival. (**a**) Percentage of BrdU-positive cells double-labelled for Ng in the adult GCL at different survival times (*n* = 3 animals at each survival time). (**b**,**c**) Density of adult-generated (6-weeks old) GCs in the deep and superficial GCL (**b**) of wt, het and Ng-ko mice. One-way ANOVA: F[2, 7] = 3.54, *p* = 0.08665 for sGCL; F[2, 7] = 0.031, *p* = 0.96937 for dGCL. *n* = 3 animals for wt and het, *n* = 4 animals for Ng-ko. dGCL: deep granule cell layer; sGCL: superficial granule cell layer.

**Figure 3 ijms-22-04269-f003:**
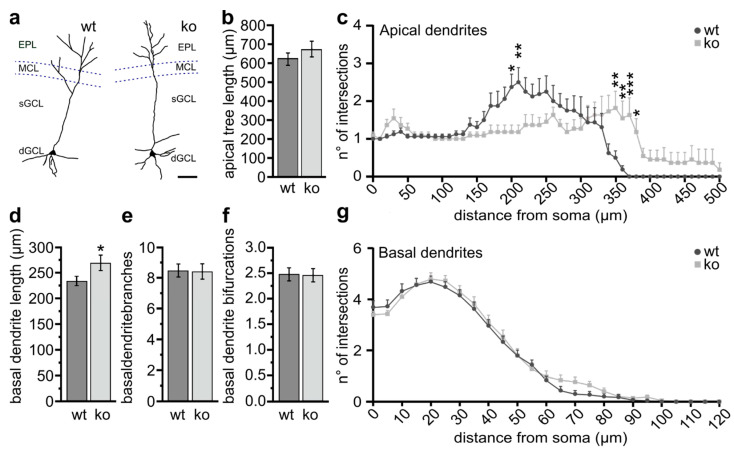
Adult-born GCs in Ng-ko mice show slightly altered dendritic morphology. (**a**) Representative images of reconstructed adult-born GFP-positive GCs in the OB of wt (left) and Ng-ko (right) mice; scale bar = 50 μm. (**b**) Analysis of GFP-positive cells apical compartment in wt and Ng-ko mice. The apical tree length does not vary between genotypes: *p* = 0.30626, Student t-test; *n* = 13 cells from 3 animals for wt and *n* = 9 cells from 3 animals for Ng-ko. (**c**) Sholl analysis on the apical dendritic extension shows increased probability of ramification between 200 and 210 μm in wt cells whereas the number of intersections increases in Ng-Ko mice between 350 and 380 μm. Data represent the mean number of intersections in 10 μm steps (*n* = 16 cells, from 3 wt animals; *n* = 11 cells, from 3 Ng-ko animals; two-way repeated measures ANOVA: genotype × distance interaction, F[50, 1250] = 3.546, *p* < 0.001; Sidak’s post-test: 200 μm: *p* = 0.03; 210 μm: *p* = 0.008; 350 μm: *p* = 0.008; 360 μm: *p* = 0.005; 370 μm: *p* < 0.001; 380 μm: *p* = 0.04). The area under curve (AUC) is 527.5 ± 46.54 in wt and 541.8 ± 42.47 in Ng-Ko animals. (**d**–**f**) Analysis of GFP-positive cells basal compartment shows that the total dendritic basal length is higher in Ng-ko compared to wt cells [(**d**): *p* = 0.03946, Student *t*-test; *n* = 40 cells from 3 animals for wt and *n* = 34 cells from 3 animals for Ng-ko] while the number of branches (**e**) and number of bifurcations (**f**) of basal dendrites do not change between the two genotypes (*p* = 0.921 for the number of branches, Student t-test, *n* = 40 cells from 3 animals for wt and *n* = 34 cells from 3 animals for Ng-ko; *p* = 0.966 for the number of bifurcations, Mann-Whitney test, *n* = 40 cells from 3 animals for wt and *n* = 34 cells from 3 animals for Ng-ko. (**g**) Sholl analysis confirms no difference in the basal dendritic arborization in wt versus Ng-ko mice. Data in g represent the mean number of intersections in 5 μm steps (*n* = 30 cells from 3 wt animals; *n* = 30 cells from 4 Ng-ko animals; two-way repeated measures ANOVA: genotype, F[1, 58] = 0.3245, *p* = 0.57; distance, F[1, 58] = 271, *p* ˂ 0.001; genotype × distance interaction, F[24, 1392] = 0.7550, *p* = 0.80). AUC: 211.6 ± 16.72 in wt and 221.7 ± 17.69 in Ng-ko mice. * *p* < 0.05; ** *p* < 0.01; *** *p* < 0.001.

**Figure 4 ijms-22-04269-f004:**
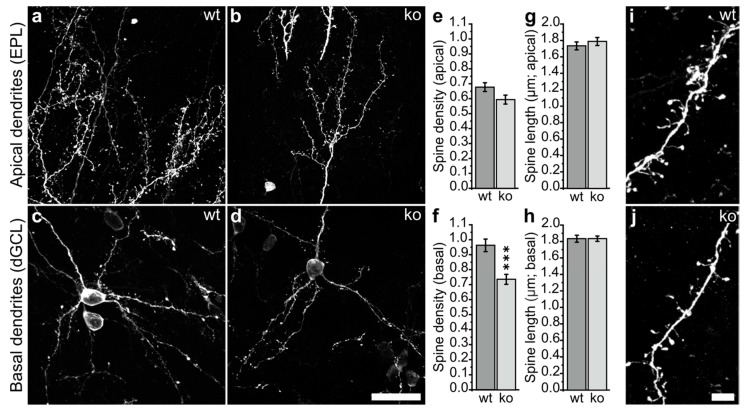
Adult-born GCs in Ng-ko mice show altered spine density. (**a**–**d**) Representative confocal images of the dendritic tree of adult-generated GFP-positive GCs in the EPL (**a**,**b**) and in the deep GCL ((**c**,**d**); dGCL) of wt and Ng-ko mice. Scale bar in d = 20 μm and applies to (**a**–**c**). (**e**,**f**) Mean spine density in the apical (**e**: *p* = 0.053, Student *t*-test; *n* = 25 dendrites from 2 animals for wt and *n* = 27 dendrites from 3 animals for Ng-ko) and basal (**f**: *p* = 4.560 × 10^−5^, Student *t*-test; *n* = 54 dendrites from 3 animals for wt and *n* = 54 dendrites from 3 animals for Ng-ko) compartments of GFP-positive GCs; (**g**,**h**) Mean spine length in apical (**g**: *p* = 0.437, Student *t*-test; *n* = 25 dendrites from 2 animals for wt and *n* = 27 dendrites from 3 animals for Ng-ko) and basal (**h**: *p* = 0.999, Student *t*-test; *n* = 54 dendrites from 3 animals for wt and *n* = 54 dendrites from 3 animals for Ng-ko) compartments of GFP-positive GCs. (**i**,**j**) Higher magnification images showing wt (**i**) and Ng-ko (**j**) basal dendritic segments with spines. Scale bar in j = 5 μm and applies to (**i**). *** *p* < 0.001.

**Figure 5 ijms-22-04269-f005:**
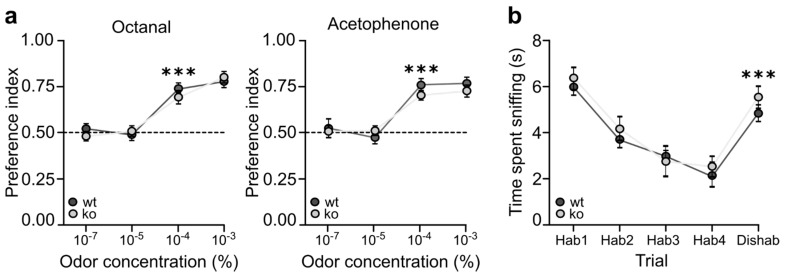
Ng-ko mice display normal odor detection and discrimination. (**a**) Odor detection threshold of wt and Ng-ko mice was calculated measuring the time they spent investigating two different odors (octanal, left and acetophenone, right) at increasing concentrations versus odorless mineral oil. Index values between 0.50 (black dotted line) and 1.00 were indicative of a preference for the odor. Effect of genotype: F[1, 19] = 2.05, *p* > 0.168; effect of trial: F[3, 44] = 27.34, *p* < 0.001; trial × genotype interaction: F[3, 44] = 0.74, *p* > 0.531; two-way repeated measures ANOVA, followed by Fisher’s PLSD post-hoc comparisons; n = 10 animals for wt and *n* = 11 animals for Ng-ko. *** *p* < 0.001 versus 10^−5^ %. (**b**) Olfactory discrimination capacity of wt and Ng-ko mice was assessed by a habituation (Hab)—dishabituation (Dishab) test. A significant increase of the sniffing time in the presence of the novel odor (Dishab) indicates normal discrimination capacity. Effect of genotype: F[1, 18] = 0.79, *p* = 0.385; effect of trial: F[4, 72] = 32.56, *p* < 0.001; trial × genotype interaction: F[4, 72] = 0.36, *p* = 0.838; two-way repeated measures ANOVA, followed by Fisher’s PLSD post-hoc comparison; n = 10 animals for wt and *n* = 10 animals for Ng-ko. *** *p* < 0.001 versus Hab-4.

**Figure 6 ijms-22-04269-f006:**
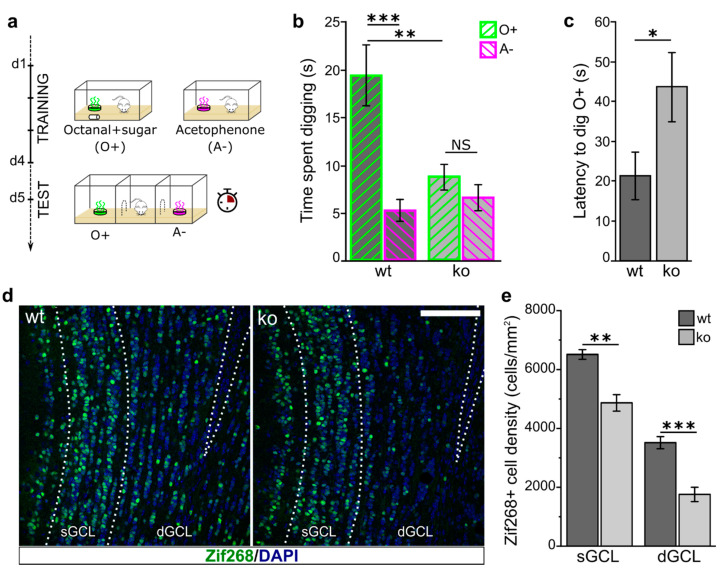
Ng-ko mice show impaired odor-reward associative memory coupled to defective Zif268 expression in GCs. (**a**) Schematic of the odor-reward associative memory test. During the training sessions (days 1–4) only octanal was associated with the sugar reward (O+) whereas acetophenone was never reinforced (A−). On day 5 (d5) both odors were presented and the time the animals spent digging the odors was recorded. (**b**) wt mice demonstrated a strong learning of the odor-reward association as shown by the time they spent digging at the reinforced odor (octanal, O+), while Ng-ko mice failed to show any significant preference for the reinforced odor. Effect of genotype: F[1, 18] = 4.33, *p* = 0.519; effect of odor: F[1, 18] = 24.86, *p* < 0.001; odor × genotype interaction: F[1, 18] = 13.36, *p* = 0.002; two-way repeated measures ANOVA, followed by Fisher’s PLSD post-hoc comparisons; *n* = 10 animals for wt and *n* = 10 animals for Ng-ko (**c**) Latency to dig calculated in the same test as in (**b**). At test, Ng-ko mice show a significantly higher latency to dig at the reinforced odor (octanal), compared to wt. *p* < 0.05; Student’s *t*-test, *n* = 10 animals for wt and *n* = 10 animals for Ng-ko. (**d**) Representative confocal images of the olfactory bulb GCL stained for Zif268 (green) and counterstained with DAPI (blue) of wt (left) and Ng-ko (right) mice sacrificed shortly after the odor-reward associative memory test (protocol in a). Scale bar = 100 µm. (**e**) Density of Zif268-positive cells in the deep and superficial GCL of wt and Ng-ko mice following the test paradigm of odor-reward associative memory (protocol in a). Ng-ko mice show a significant reduction of Zif268-positive cells in both deep and superficial GCL: *p* = 0.0012 for sGCL, Student *t*-test; *n* = 5 animals for wt and *n* = 4 animals for Ng-ko; *p* = 9.236 × 10^−4^ for dGCL, Student *t*-test; *n* = 5 animals for wt and *n* = 4 animals for Ng-ko. dGCL: deep granule cell layer; sGCL: superficial granule cell layer. * *p* < 0.05; ** *p* < 0.01; *** *p* < 0.001; NS: not significant.

## Data Availability

The data that support the findings of this study are available from the corresponding author upon request.

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
