# Peer review of "Neurogranin Regulates Adult-Born Olfactory Granule Cell Spine Density and Odor-Reward Associative Memory in Mice"

_ijms, 2021, doi:10.3390/ijms22084269_

Round 1

Reviewer 1 Report

The authors investigated how Neurogranin knockout changes the olfactory bulb synaptic plasticity and odor-reward associative function. By using Ng knock-out (Ng-ko) mice, the authors analyzed the neurogenesis, dendritic morphology and spine density of adult born granule cells. Furthermore, odor-reward associative memory was tested. I have some concerns about experiment design.

Major comments:

  1. GABAergic olfactory GC interneurons were investigated in this study. It is well-known that excitatory neurons carry spines, whereas inhibitory neurons do not. This rule has its exceptions. Could the authors give evidence to confirm 1) whether or not the neurons they analyzed were GABAergic neurons 2) Do olfactory GC interneurons have spines?
  2.  Odor-reward associative memory experiment. Did the mice have natural preference for octanal? Sugar should randomly accompany with octanal or acetophenone. 
  3. Ng knock-out (Ng-ko) mice is whole-brain knockout, could it possible that other brain regions are involved in the odor-reward associative memory dysfunction? 

Author Response

We thank the reviewer for the revision of our study. A detailed point-to-point answer to its major concerns is presented in the attached pdf.

Reviewer 2 Report

The authors here present an excellent manuscript detailing an investigation of the role of Neuogranin in regulating adul-born olfactory granule cell spine density and odor -reward associate memory in a mouse model. The authors have demonstrated a well rounded design of experiments and have shown through the use of a knockout model supporting information for the role of neurogranin in molecular mechanisms that underly granule cell plasticity.

Overall, i believe in its current status the article is suitable for publication, 

Author Response

We thank the reviewer for the very positive feedback about our work.

Best Regards

Round 2

Reviewer 1 Report

The authors adequately explained my concerns.